# How Transferable are Video Representations Based on Synthetic Data?

**Yo-whan Kim[1,2], Samarth Mishra[2,4], SouYoung Jin[1],**
**Rameswar Panda[2], Hilde Kuehne[2,3], Leonid Karlinsky[2],**
**Venkatesh Saligrama[4], Kate Saenko[2,4], Aude Oliva[1,2], Rogerio Feris[2]**

[1]MIT, [2]MIT-IBM Watson AI Lab, [3]Goethe University, [4]Boston University

## Abstract

Action recognition has improved dramatically with massive-scale video datasets. Yet, these datasets are accompanied with issues related to curation cost, privacy, ethics, bias, and copyright. Compared to that, only minor efforts have been devoted toward exploring the potential of synthetic video data. In this work, as a stepping stone towards addressing these shortcomings, we study the transferability of video representations learned solely from synthetically-generated video clips, instead of real data. We propose SynAPT, a novel benchmark for action recognition based on a combination of existing synthetic datasets, in which a model is pre-trained on synthetic videos rendered by various graphics simulators, and then transferred to a set of downstream action recognition datasets, containing different categories than the synthetic data. We provide an extensive baseline analysis on SynAPT revealing that the simulation-to-real gap is minor for datasets with low object and scene bias, where models pre-trained with synthetic data even outperform their real data counterparts. We posit that the gap between real and synthetic action representations can be attributed to contextual bias and static objects related to the action, instead of the temporal dynamics of the action itself. The SynAPT benchmark is available at https://github.com/mintjohnkim/SynAPT.

## 1   Introduction

Large-scale pre-training using massive datasets, containing hundreds of thousands or even millions of video clips, have brought significant progress in action recognition [28, 38, 1, 17]. High capacity models trained on such large datasets have shown remarkable generalization performance to downstream tasks where training data is limited [64, 17].

While this progress is exciting, these large-scale video datasets also have shortcomings. Collecting and annotating videos is expensive, tedious, and time-consuming. As a result, methods that learn feature representations from unlabeled videos, including self-supervised [61], weakly-supervised [17], and semi-supervised approaches [53], have received significant attention in recent years. However, these works do not address other important ethical, legal, and technical issues related to processing real-world data, as described below:

**Privacy concerns.** Video samples may include human interactions and activities, but often, the individuals' sensitive information (e.g., faces, license plates, or location indicators) is captured along.

**Proprietary issues.** Massive-scale datasets containing millions or billions of images and videos, such as IG65M [17] and JFT3B [65], are not publicly available, preventing the larger community from reproducing results, which hinders research progress.

36th Conference on Neural Information Processing Systems (NeurIPS 2022) Track on Datasets and Benchmarks.

**Pre-training: Synthetic Videos**

Pointing with finger | Fall to the floor | Nausea vomiting | Waving | Walk hold hands | Pull up

**Downstream Tasks**

Decreasing scene-object bias

UCF101 | HMDB51 | Mini-SSV2 | Diving48 | IkeaFA | UAV-Human

playing violin | ride horse | spilling something next to something | reverse dive no-twist tuck | flip table | dig a hole

Figure 1: We introduce SynAPT, a novel action recognition benchmark, to pre-train a model on synthetic videos and transfer learned knowledge to real downstream tasks with label sets disjoint from pre-training data. We observe that the models pre-trained on synthetic videos can even outperform those pre-trained on real videos when the downstream datasets have low object and scene bias.

**Ethical issues and bias.** Ethical issues related to skin tone and gender [6], and unwanted contextual bias are difficult to control in existing large-scale datasets. Consequently, state-of-the-art models may fail to predict actions such as a person dancing in a mall [11], or a woman snowboarding [21].

**Data protection and copyright issues.** Data collected without consent, which is common for existing massive-scale datasets, may violate copyright as well as data protection laws such as the General Data Protection Regulation (GDPR).

A promising way to address these issues is using computer-generated synthetic videos for pre-training. By leveraging 3D models of humans and scenes, an arbitrarily large number of videos can be generated by varying simulation parameters such as lighting, texture, and background, while enabling the control of sensitive attributes of humans, such as gender and race. This approach of training with synthetic data has a long history in computer vision [15, 33, 39]. Recent efforts on action recognition [23, 45, 58] have used completely synthetic datasets or synthetic/real hybrid datasets to train deep neural network models. However, these works rely on domain adaptation techniques that assume *the same label set* for both synthetic data and real data. This might not always be feasible as each action class requires motion capture or simulation capacities. To the best of our knowledge, no previous work has studied the transferability of action representations based on synthetic data to diverse downstream tasks, where the synthetic and real domains have disjoint label sets.

In this work, we introduce a novel benchmark named Synthetic Action Pre-training and Transfer (SynAPT), that addresses this important problem. As shown in Figure 1, our pre-training dataset consists solely of synthetic video clips. Based on existing synthetic datasets [23, 45, 58], we compile a new benchmark with 150 action categories, where each category has 1,000 samples. We consider six downstream datasets: UCF101 [54], HMDB51 [30], Something-Something V2 [18], Diving48 [32], Ikea Furniture Assembly (IkeaFA) [55], and UAV-Human [31]. Both UCF101 and HMDB51 datasets are based on YouTube videos depicting a broad variety of actions. As a result, they exhibit a high object and scene bias [32], i.e., several actions can be recognized by just looking at static objects or the background, as opposed to the action itself. For example, the action "playing violin" could likely be recognized by detecting the object violin instead of understanding the temporal dynamics

of the action. On the other hand, the remaining four datasets have low object and scene bias, as understanding temporal dynamics is needed to correctly recognize actions in these datasets.

Based on this setting, we conducted an extensive analysis on the transferability of pre-trained video models based on synthetic data, including the effect of linear probing and fine-tuning, number of classes, number of samples per class, and their relative performance with respect to pre-trained ImageNet models, which are used to measure the object and scene representation bias of each dataset. We solidify our findings by replicating the experiments with models of various capacities and complexity, and further perform rigorous hyperparameter sweeping on the downstream tasks.

It shows that the transferability gap between synthetic and real action recognition models is directly related to the object and scene bias of the datasets. Models pre-trained on Kinetics clearly outperform synthetic pre-trained models on datasets with high bias (UCF101, HMDB51). The gap is closed for datasets with low bias (Mini-SSV2, Diving48, IkeaFA, UAV-Human), where synthetic pre-trained models achieve similar or better accuracy than their real counterparts.

In summary, the main contributions of this work are as follows:

1. We propose a novel benchmark, SynAPT, for studying the transferability of synthetic video representations for action recognition. To the best of our knowledge, no previous work has investigated this problem before.

2. We extensively study the transferability of synthetic video representations for action recognition, showing that the simulation to real gap is simply closed for datasets with low object and scene bias, but still exists for datasets with high bias. This result suggests that the gap between real and synthetic action representations exists largely due to contextual bias and static objects related to the action, instead of the temporal dynamics of the action itself.

We hope that the proposed benchmark provides a direction to mitigate ethical and legal issues with existing large-scale datasets of real images and videos for action recognition research.

## 2 Related Work

**Action Recognition Benchmarks.** Video datasets have rapidly evolved from small-scale benchmarks such as KTH [50] and Weizmann [5], with a few thousand video clips, to medium-scale datasets such as UCF101 [54] and HMDB51 [30], and recently to large-scale datasets containing hundreds of thousands or millions of annotated videos, such as Kinetics [28], YouTube 8M [1], and the Moments in Time dataset [38]. It is well-established that pre-training on such large datasets followed by fine-tuning on downstream tasks boosts performance, especially when the target datasets are small [54, 30, 25, 63, 19, 18, 52]. With the challenges of curating and defining label taxonomies for massive-scale datasets, the focus has shifted to pre-training on unlabeled videos [61, 53], or video datasets accompanied by weak supervision such as social media hashtags [17] or narrated instructions [35], which can be obtained without expensive data curation. Compared to existing action recognition benchmarks that use real-world datasets, we propose a novel benchmark that aims at studying *pre-training and transfer from synthetic videos*, as a stepping stone to mitigate issues related to privacy, bias, ethics, and copyright.

**Learning from Synthetic Data.** Synthetic data has been widely used to solve various computer vision problems by replacing real-world training data [15, 36, 40, 42, 46, 60, 45, 58, 26, 29, 49, 62]. While many of these works have tried to generate synthetic data as similar as real data, Baradad et al. [2] has shown that synthetic images with structured noise can be used for representation learning as the diversity of training images is as important as naturalism. Further, approaches to optimizing simulator parameters have been explored to learn better synthetic data for specific tasks [3, 48, 27], or even tasks not seen during training [37].

Only a few works attempted to learn action recognition from synthetic data. ElderSim [23] generates realistic videos of elders' daily activities in households to augment limited publicly available elder activity data. SURREACT [58] introduced a novel data generation method that reconstructs 3D human body models from videos to render synthetic videos from unseen viewpoints at various angles. Procedural Human Action Videos dataset (PHAV) [45] generates human action videos that show more overlap with traditional dataset classes such as "push", "kick ball", "walking", or "hug". It combines existing motion capture sequences with synthetic generated actions based on physically

plausible motion variations. In our work, *we compile a large-scale synthetic video dataset* to explore a mixture of these simulators as a synthetic pre-training baseline for video backbones in general.

Existing approaches to use simulators for action recognition [15, 23, 58] have shown performance improvement by adding the simulated videos to the original training datasets. However, in contrast to our proposed benchmark, no prior work has studied the transferability of synthetic video representations to other domains that may have different action categories than the synthetic datasets.

**Domain Knowledge Transfer from Synthetic Data.** Many approaches have been proposed to transfer knowledge from synthetic to real domains, generally relying on standard domain adaptation methods [12, 34, 13, 10] to bridge the gap between the two domains. Examples include generative models to improve the realism of synthetic images and videos [44, 22], as well as methods that operate in the feature space, such as adversarial methods which encourage domain confusion to learn domain-invariant features [43, 16, 57], and discrepancy-based approaches that align feature distributions of the two domains [47, 66, 9]. More recently, Syn2Real [41], a large-scale synthetic-to-real benchmark has been introduced for unsupervised domain adaptation.

These domain adaptation methods assume the same label set between the synthetic and real domains. By contrast, in our work, we remove this assumption and instead consider *multiple downstream tasks with disjoint label sets.* In addition, while prior work has been focused on adapting video representations from the synthetic to real domains, we show that the action recognition performance gap between these domains is directly related to the object and scene bias of the downstream datasets.

## 3 Proposed Benchmark

As discussed in the introduction, synthetic videos can help avoid many issues that accompany real videos, for use in prediction tasks. Towards this end, we propose SynAPT, a novel benchmark, consisting of a dataset of only synthetically generated video clips, curated using publicly available assets: 1) ElderSim [23], 2) SURREACT [58], and 3) PHAV [45]. While these assets have been used previously for recognition tasks within the same label sets, we use them to pre-train video backbones, which we subsequently transfer to downstream datasets with label sets different from synthetic data.

### 3.1 Synthetic Dataset Sources

**ElderSim** [23] is a dataset based on videos of elders' daily activities in households, along with 2D and 3D skeleton trajectories, with a goal of augmenting limited publicly available elder activity data. ElderSim has four realistic 3D rendered furnished residential house models for background with flexible lighting and camera viewpoint options. There are 15 different human agents with various simulation parameters, such as skin color, outfits and gender. Overall, the dataset features 462K videos based on 55 different action classes captured under 28 viewpoints. Classes include daily activities such as "eating food with a fork" or "sitting up/standing up".

**SURREACT** [58] reconstructs 3D human body models from videos to render synthetic videos for unseen viewpoints at various angles. For this work, we use the SURREACT data based on the body pose information of two datasets, UESTC [24] and NTU [51]. Note that compared to ElderSim, SURREACT only supports static images as background. For each original sample, eight different synthetic videos are generated with varying viewpoints, human body shape, clothes, and gender.

**Procedural Human Action Videos (PHAV)** [45] is a large scale synthetic dataset generated using modern game engines, providing physically plausible motions and actions. PHAV contains actions performed by 20 artist-designed human models at seven different large-scale environment backgrounds. The videos have four lighting settings based on period of day, as well as four weather options. Around 40,000 videos are provided, with at least 1,000 examples per class.

### 3.2 Synthetic Dataset Curation

Using the generators/dataset described in Section 3.1, we created our Synthetic dataset with 150 classes in total. 55 actions from ElderSim, 100 actions from SURREACT, and 35 actions from PHAV were collected. We manually screened and combined overlapping classes, and randomly selected 1,000 samples for each class, resulting in a collection of 150,000 videos. For classes with samples from multiple assets, an equal number of videos were sampled from each asset to maintain

an adequate ratio. We extracted frames at a constant frame rate of 30 frames per second. We will provide the respective scripts for the community to generate this synthetic data.

## 3.3 Downstream Tasks

To assess the transferablity of video representations based on synthetic data, we fine-tuned and linear probed the pre-trained models on six different downstream tasks. In this subsection, we describe the details of the datasets used for the downstream tasks.

**UCF101** [54] is a human-action dataset collected from YouTube, consisting of 101 action classes with 13,320 videos in total. UCF101 contains various realistic action classes, as well as subdivided organization methodology (i.e. action categories are further divided into five types and 25 groups, in which videos in a same group have common qualities such as background or viewpoint).

**HMDB51** [30] presents 51 human activities with refined quality, light conditions, and accurate surrounding features, and is thus smaller than UCF101 with only 6,849 clips. HMDB51 is further divided into five types, including rather detailed action classes such as "smiling" or "laughing".

Something-Something V2 [18] was introduced to test the ability of a model to understand temporal dynamics rather than relying on objects or background in scenes. The dataset consists of 174 classes with around 220,000 videos of humans performing basic actions with common objects, in which action labels are independent of the objects themselves (e.g. "putting something behind something"). For our experiments, we use a reduced version of this dataset named **Mini-SSV2** [8], which consists of only half of the action labels. 87 labels are chosen at random, resulting in around 93,000 videos.

**Diving48** [32] is a collection of diving competition videos, made up of around 18,000 videos which are divided into 48 dive sequences. Since all videos share a similar background and object features, Diving48 is considered a fine-grained dataset and is often used to test the robustness of video models.

**Ikea Furniture Assembly** [55], or IkeaFA, provides 111 videos, each 2-4 minutes long. Summing up to around 480,000 frames of data, IkeaFA is a collection of GoPro furniture assembly videos, all of which are collected under a constant background by 14 individuals, either on a table or on the floor. There are 12 action classes in IkeaFA, including "pick leg", "attach leg", and "flip table".

**UAV-Human** [31] dataset is collected using an Unmanned Aerial Vehicle, thus providing a collection of videos from unique viewpoints. The dataset provides different recording modalities (i.e. fisheye videos and night-vision videos). To stay consistent with the pre-training and other downstream datasets, we only utilize videos based on the standard RGB camera for this work. Those comprise 22,476 videos depicting 155 action classes, collected from 119 subjects. Note that for all reported numbers in the following sections, we use cross-subject-v1 evaluation method as described in [31].

# 4 Experiments

## 4.1 Implementation Details with Various Model Architectures

We ran experiments on three different model architectures with various capacities and complexity: Temporal Segment Network (TSN) [59], I3D [7], and R(2+1)D [56], covering 2D, 3D, and 2.5D feature representations, respectively. We use ResNet-50 [20] backbone for all the models. Note that we trained all models in this paper from scratch without ImageNet pre-trained weights unless specified. Refer to Appendix A for a description of the three architectures. Pre-trained model weights are available at https://github.com/mintjohnkim/SynAPT.

**Hyperparameter Sweeping.** The Kinetics and Synthetic baseline models for TSN, I3D, and R(2+1)D were trained using SGD optimizer with momentum of 0.9, final layer dropout rate of 0.5, and number of samples of 8 frames per clip. We examined initial learning rates [0.01, 0.02] with cosine decay, batch sizes [64, 128], and weight decay rates [0.0001, 0.0005, 0.001], resulting in total 12 combinations of hyperparameters per baseline model. We selected the best-performing model for each baseline to transfer onto our downstream tasks.

For each downstream task, while keeping the optimizer, momentum, dropout rate, and number of samples fixed, we explored initial learning rates [0.0001, 0.0005, 0.001], batch sizes [32, 64], and weight decay rates [0.0001, 0.0005, 0.001], resulting in total 18 hyperparameter combinations per downstream task per baseline model.

Table 1: Top-1 accuracy results via fine-tuning (FT) and linear probing (LP) on downstream tasks.

| Model | Pre-training Dataset | Downstream Dataset | | | | | | | | | | | |
|---|---|---|---|---|---|---|---|---|---|---|---|---|---|
| | | UCF101 | | HMDB51 | | Mini-SSV2 | | Diving48 | | IkeaFA | | UAV-Human | |
| | | FT | LP | FT | LP | FT | LP | FT | LP | FT | LP | FT | LP |
| TSN | Kinetics | **86.17** | **69.92** | **57.45** | **46.34** | 48.50 | 8.61 | 62.84 | 9.65 | 42.07 | 31.71 | 32.45 | 3.52 |
| | Synthetic | 83.40 | 28.02 | 54.38 | 20.92 | **49.69** | **12.82** | **63.50** | **10.91** | **42.68** | **35.98** | **35.57** | **5.70** |
| | Scratch | 48.37 | | 20.59 | | 39.58 | | 15.94 | | 32.32 | | 2.18 | |
| I3D | Kinetics | **86.87** | **68.09** | **59.21** | **46.13** | 50.08 | 8.56 | 54.82 | 9.09 | 40.85 | 32.67 | 31.13 | 3.15 |
| | Synthetic | 82.05 | 27.57 | 55.69 | 22.59 | **50.72** | **12.31** | **55.28** | **10.10** | **42.68** | **33.22** | **35.13** | **5.83** |
| | Scratch | 46.37 | | 18.82 | | 39.77 | | 12.64 | | 34.76 | | 1.97 | |
| R(2+1)D | Kinetics | **87.21** | **69.44** | **58.33** | **47.19** | 51.48 | 8.76 | 53.04 | 9.80 | 39.02 | 34.49 | 29.83 | 3.52 |
| | Synthetic | 80.02 | 26.41 | 53.27 | 22.22 | **52.01** | **13.26** | **57.31** | **10.00** | **41.46** | **35.74** | **31.79** | **5.49** |
| | Scratch | 42.45 | | 16.86 | | 39.07 | | 11.87 | | 31.10 | | 1.26 | |

Table 2: Representation bias for each pre-training and downstream dataset computed using TSN models. (LP = linear probing. Please refer to Appendix B for the representation biases computed using I3D and R(2+1)D models.)

| | Pre-training | | Downstream Dataset | | | | | |
|---|---|---|---|---|---|---|---|---|
| | Kinetics | Synthetic | UCF101 | HMDB51 | Mini-SSV2 | Diving48 | IkeaFA | UAV-Human |
| ImageNet LP Accuracy, $\mathcal{M}(\mathcal{D}, \phi)$ | 44.67 | 22.32 | 65.32 | 38.56 | 13.68 | 10.96 | 34.76 | 2.59 |
| Representation Bias, $\mathcal{B}(\mathcal{D}, \phi)$ | 6.07 | 5.07 | 6.04 | 4.30 | 3.57 | 2.40 | 2.06 | 2.01 |

## 4.2 Transfer Learning Results

We first present the transfer learning experiments by showing finetuned and linear probing top-1 accuracy for backbones pre-trained on Kinetics, Synthetic, as well as with random initialization (Scratch) in Table 1. Note that the Synthetic dataset used for pre-training consists of 150 classes with 1000 samples per class, and the Kinetics dataset used for pre-training is down-scaled to match the Synthetic dataset's statistics. All classes and samples for the downsized Kinetics dataset were randomly selected from full Kinetics [28]. We emphasize that our goal is not to obtain state-of-the-art results on the downstream datasets, given the reduced pre-training dataset sizes as described above. Instead, we aim at providing a fair comparison between real and synthetic models, that will allow researchers to operate with synthetic data in a way that it is comparable to real-world baselines. We show both fine-tuning and linear probing transfer results.

While the Kinetics pre-trained model is preferable for UCF101 and HMDB51, our Synthetic pre-trained model outperforms the Kinetics model when transferring on Mini-SSV2, Diving48, IkeaFA, and UAV-Human. Qualitatively, we conjecture that UCF101 and HMDB51 are more prone to object and scene representation bias than the other four datasets. We further assume that the Synthetic dataset is more robust to bias than Kinetics since clips are generated on either shared background image/rendering or without surrounding objects in relation to the action class, which forces the model to focus on the actions over possible biases.

To further analyze this property of generated synthetic videos, we quantify representation bias for each downstream dataset using the following equation by borrowing the definition from [32]:

$$\mathcal{B}(\mathcal{D}, \phi) = \log \frac{\mathcal{M}(\mathcal{D}, \phi)}{\mathcal{M}_{rnd}}, \tag{1}$$

where bias $\mathcal{B}$ for dataset $\mathcal{D}$ using representation $\phi$ is directly related to the ratio of the performance of subject representation $\mathcal{M}(\mathcal{D}, \phi)$ to random chance performance, $\mathcal{M}_{rnd}$. We calculate $\mathcal{M}(\mathcal{D}, \phi)$ by measuring the performance of a linear action recognition classifier trained on top of a frozen ImageNet model. The intuition is that ImageNet features encode static cues, such as objects, and therefore $\mathcal{M}(\mathcal{D}, \phi)$ is related to the amount of action categories that can be recognized solely by static cues in the videos, without any temporal dynamics.

Table 2 summarizes the representation bias measured using an ImageNet pre-trained model for downstream tasks. As hypothesized, UCF101 and HMDB51 have high representation bias scores of

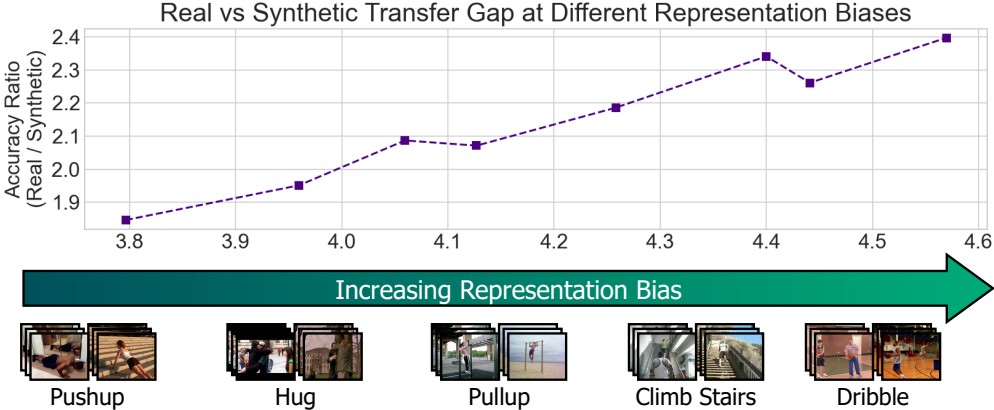

Figure 2: Representation Bias vs Transferability. We see the downstream performance gap in Synthetic vs Real data pre-training reduces with smaller downstream representation bias. The different points on the x-axis correspond to different 25-class subsets of the HMDB dataset. Shown below are examples of classes in increasing order of bias—higher bias subsets are constituted of classes more biased towards scenes and objects.

6.04 and 4.30, respectively, while Mini-SSV2, Diving48, IkeaFA, and UAV-Human have much lower representation bias scores of 3.57, 2.40, 2.06, and 2.01, respectively. It is expected that UCF101 and HMDB51 have high biases as they are composed of daily human actions with related objects and scene features in them. In addition, while Mini-SSV2 shows lower bias score than UCF101 and HMDB51 as its action categories focus on temporal movement/change of objects rather than objects themselves, it still has higher bias than Diving48 as models can learn unintentional object bias since some objects are more prone to specific action categories than others (e.g. round objects are more inclined to roll). Every IkeaFA video is taken under the same setting, with identical objects present in the frame throughout the entire video, and this consistency is reflected by its low representation bias. Finally, UAV-Human also exhibits low representation bias as UAV's far-distance viewpoint encompasses vast scene and object information, degrading the model's ability to predict an action based on such information.

We observe that pre-training with Kinetics tends to perform better on downstream datasets with high representation bias, while pre-training with Synthetic data performs better on tasks with lower representation bias. Looking at the biases of the pre-training datasets themselves, we see Kinetics has a much higher representation bias than Synthetic. This suggests that pre-training on datasets with low bias, which can be achieved more easily with synthetic data, can perform well on downstream tasks with low bias, and vice-versa.

### 4.2.1 Sensitivity Analysis

We conduct various sensitivity analyses to capture the transferability of video representations learned from the Synthetic data. Here we show results using TSN [59]. Please refer to Appendix C and D for additional results with I3D [7] and R(2+1)D [56].

**Representation Bias and Transferability**   To further investigate the inverse relation between representation bias and the transfer performance gap, we conducted the following experiment. We ranked the classes of HMDB51 by difficulty based on average accuracy over its train examples of a linear probe attached to an ImageNet pre-trained model. Hence, a dataset with less difficult classes is expected to have high ImageNet representation bias. This way, we created 8 different 25-class subsets of HMDB-51 at different representation biases. Figure 2 shows the downstream performance of linear probes (reported as a ratio) for real vs synthetic pre-training on these different subsets. We see, similar to our observation in Section 4.2, that as the representation bias of the downstream data decreases, so does the performance gap of real vs. synthetic pre-training.

**Effect of Number of Classes**   We study how the number of classes in pre-training datasets influences the transferability on our downstream tasks, and analyze its relationship with representation bias. For both Kinetics and Synthetic, we created three datasets with 30, 90, and 150 classes respectively, all

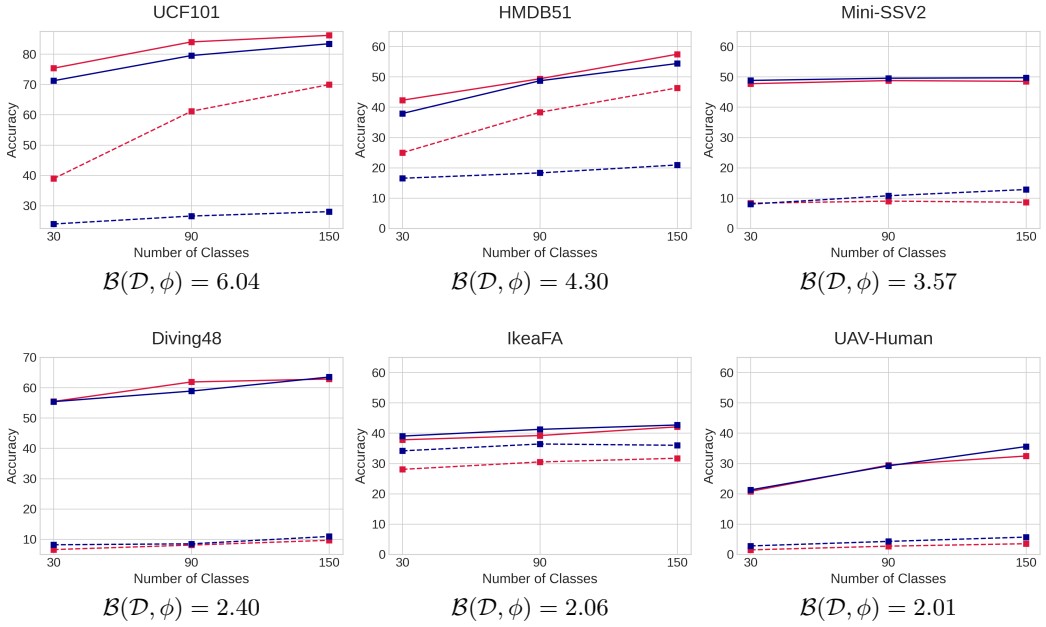

Figure 3: — Kinetics FT; - - - Kinetics LP; — Synthetic FT; - - - Synthetic LP. Fine-tuning (FT) and Linear Probing (LP) transfer results on six downstream tasks with various number of classes in pre-training datasets. We use TSN ResNet-50 [59] for the experiments above. Best viewed in color.

with 1000 samples per class. Note that subset classes were chosen randomly, and each dataset created is a superset of every other smaller dataset.

Figure 3 plots the fine-tuning and linear probing transfer top-1 accuracies of all pre-trained models on downstream tasks. Representation bias and pre-training datasets' human-motion bias explain the trends above. For the high representation bias datasets, i.e. UCF101 and HMDB51, we see a significant gain in accuracy as we increase the number of pre-training classes. These datasets consist of human action classes, which conceptually overlap with both the Kinetics and Synthetic actions, and hence, enjoy the extra discriminative features learned from additional classes introduced during pre-training.

On the other hand, Mini-SSV2 is not a human-action task, while Diving48 is composed of fast-paced fine-grained human movement action classes. These two low representation bias tasks therefore have low human-motion bias, and as a result, we do not see much improvement in accuracy with more pre-training classes. Although IkeaFA and UAV-Human have low representation bias, both are human action tasks with higher human-motion bias. While IkeaFA shows a marginal improvement, UAV-Human shows a more dramatic increase in accuracy. We note that unlike other low representation bias datasets, UAV-Human has some classes overlapping with pre-training datasets (see Appendix F) and it is the most coarse grained actions among the four low representation bias datasets.

**Effect of Samples per Class** We varied the number of samples per class for Kinetics and Synthetic pre-train datasets to examine its effects on transferability. We fixed the number of classes to 150, and created three subsets with 250, 750, and 1000 samples per class, with samples being chosen at random and each dataset being a superset of every other smaller dataset.

Generally, we detect a slight increase in accuracy as we increase the number of samples per class due to more availability of pre-training samples. However, we observe in Figure 4 that increasing samples per class does not significantly boost transferability compared to increasing the number of classes as we are less likely to introduce novel representation bias with extra samples within the same class. Similarly, although increasing the number of synthetic samples implies further variation in lighting, camera angles/position, humanoid types, and other video generation parameters, it does not deliver striking performance enhancement as it is not addressing the representation bias issue.

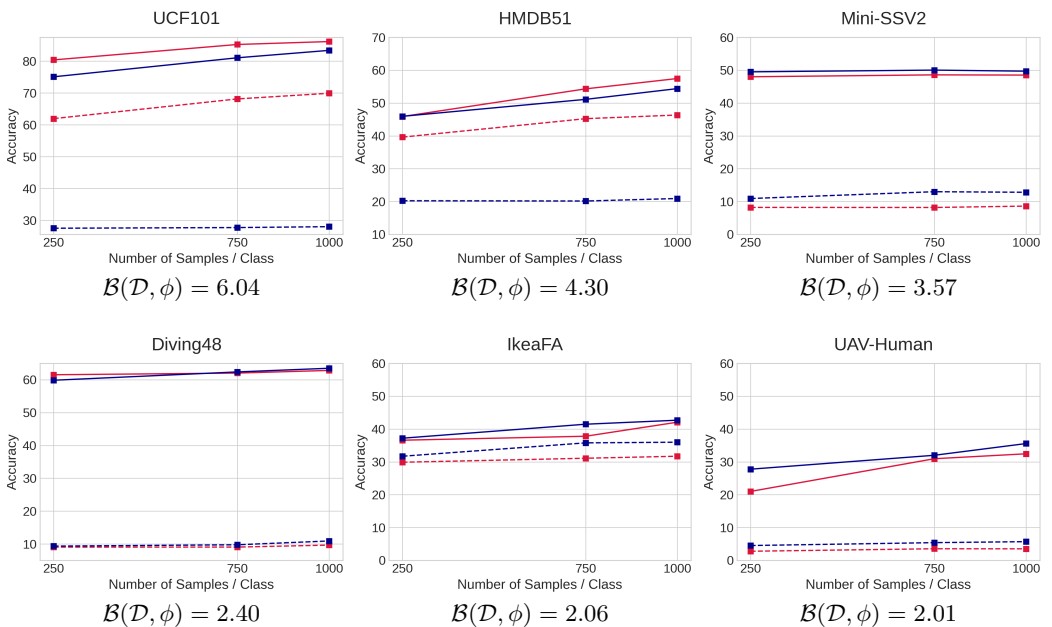

Figure 4: — Kinetics FT; - - - Kinetics LP; — Synthetic FT; - - - Synthetic LP. Fine-tuning (FT) and Linear Probing (LP) transfer results on six downstream tasks with various number of samples per class in pre-training datasets. In general, the transferability does not significantly boost compared to increasing the number of classes. Best viewed in color.

Table 3: Baseline with TimeSformer [4] (with divided space-time attention) transformer backbone.

| Model | Pre-trained Dataset | Downstream Dataset | | | | | | | | | | | |
|---|---|---|---|---|---|---|---|---|---|---|---|---|---|
| | | UCF101 | | HMDB51 | | Mini-SSV2 | | Diving48 | | IkeaFA | | UAV-Human | |
| | | FT | LP | FT | LP | FT | LP | FT | LP | FT | LP | FT | LP |
| TimeSformer | Kinetics | **92.09** | **89.43** | **59.51** | **55.42** | 48.92 | **21.54** | **46.37** | 16.99 | 61.93 | **47.73** | 23.27 | 8.40 |
| | Synthetic | 89.03 | 82.12 | 54.38 | 49.17 | **51.10** | 21.24 | 44.86 | **19.20** | **63.64** | 45.45 | **24.95** | **13.79** |

### 4.2.2 Additional Results

**Transformer Baseline.** Besides the baselines used in Table 1 which were all based on a ResNet-50 backbone, we experimented with TimeSformer (divided space-time variant) [4] which uses a transformer backbone. The results are reported in Table 3. Similar to other baseline methods, we can see the downstream performance on tasks with higher bias is better with Kinetics pre-training than with Synthetic pre-training and vice-versa for lower bias downstream tasks. We note here that unlike other baselines in Table 1, this backbone is pre-trained on ImageNet-21K as is common in the use of transformers for action recognition [14, 4].

**Using a Different Real Video Dataset for Pre-training.** To find if using another real video dataset for pre-training shows similar downstream behavior as Kinetics, we pre-trained the same TimeSformer model (with divided space-time attention) on a 150 class subset of Moment-in-Time (MiT) dataset [38]. From the results in Table 4, we find that the behavior is similar to Kinetics pre-training in that downstream performance is high on high bias downstream tasks and vice-versa.

**16-frame Input Models.** To study how changing the number of input frames affects our results, we pre-trained 16-frame TSN models using the 150 classes Kinetics and Synthetic dataset, and then evaluated them on the downstream tasks with full network fine-tuning (FT). From the results in Table 5, we find that the model generally performs better with 16 frames than with 8 frames. Additionally, similar to the 8-frame model, Kinetics pre-training performs better than Synthetic on downstream tasks with high representation bias, and opposite happens on tasks with low bias.

For more experiments, including sensitivity analyses with different baseline models and details about the benchmark, including downstream task statistics, please refer to the Appendix.

Table 4: Pre-training with Moments-in-Time [38] as a different real video dataset than Kinetics.

| Model | Pre-trained Dataset | Downstream Dataset | | | | | | | | | | | |
|---|---|---|---|---|---|---|---|---|---|---|---|---|---|
| | | UCF101 | | HMDB51 | | Mini-SSV2 | | Diving48 | | IkeaFA | | UAV-Human | |
| | | FT | LP | FT | LP | FT | LP | FT | LP | FT | LP | FT | LP |
| TimeSformer | MiT | **91.24** | **87.95** | **57.01** | **52.71** | 48.15 | 20.27 | **45.16** | 14.87 | 46.59 | 42.05 | 21.67 | 7.15 |
| | Synthetic | 89.03 | 82.12 | 54.38 | 49.17 | **51.10** | **21.24** | 44.86 | **19.20** | **63.64** | **45.45** | **24.95** | **13.79** |

Table 5: Pre-training and fine-tuning a 16-frame TSN model. Using more frames does not change the trends observed with Synthetic vs Kinetics pre-training in Sec. 4.2

| Model | Pre-trained Dataset | Downstream Dataset | | | | | |
|---|---|---|---|---|---|---|---|
| | | UCF101 | HMDB51 | Mini-SSV2 | Diving48 | IkeaFA | UAV-Human |
| TSN | Kinetics | **87.02** | **60.52** | 50.14 | 68.38 | 38.41 | 33.82 |
| (16-frames) | Synthetic | 86.10 | 59.61 | **53.68** | **70.30** | **40.24** | **40.51** |

# 5 Conclusions

In this paper, we introduced SynAPT, a new action recognition benchmark, to mitigate the issues inherent to training models with real videos, such as privacy, bias, ethics, and copyright. Specifically, we constructed a Synthetic dataset from three publicly available assets (ElderSim, SURREACT, PHAV), trained models on the Synthetic dataset, and finally transferred these pre-trained models to various downstream tasks. Our experiments show that the models pre-trained on the Synthetic dataset outperform those pre-trained on real videos on the downstream datasets with low representation bias (Mini-SSV2, Diving48, IkeaFA, UAV-Human). This suggests that although models trained on synthetic data expose weaker object and background scene features, they do provide features with strong correlation to actions, making them more useful for downstream tasks with lower representation bias. In fact, stronger object features (inherent to models trained with real videos) may even be a nuisance factor for transfer tasks onto lower representation bias datasets. We believe SynAPT and the in-depth analysis of models pre-trained on synthetic data will motivate their wider use, helping avoid different issues with real data including those related to ethics and bias.

# 6 Acknowledgements

This material is based upon work supported by the Defense Advanced Research Projects Agency (DARPA) under Contract No. FA8750-19-C-1001. Any opinions, findings and conclusions or recommendations expressed in this material are those of the author(s) and do not necessarily reflect the views of the Defense Advanced Research Projects Agency (DARPA). This work is also supported by the MIT-IBM Watson AI Lab and its member companies, Nexplore and Woodside.

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
