# Supplementary Materials: How Transferable are Video Representations Based on Synthetic Data?

**Yo-whan Kim[1,2], Samarth Mishra[2,4], SouYoung Jin[1],**
**Rameswar Panda[2], Hilde Kuehne[2,3], Leonid Karlinsky[2],**
**Venkatesh Saligrama[4], Kate Saenko[2,4], Aude Oliva[1,2], Rogerio Feris[2]**

[1]MIT, [2]MIT-IBM Watson AI Lab, [3]Goethe University, [4]Boston University

## A    Model Architecture Descriptions

**TSN:** Temporal Segment Network (TSN) [9] is an efficient 2-Dimensional Convolution Neural Network architecture designed for action recognition tasks especially with limited training samples. TSN aims to model long-range temporal structure by dividing a video input into $K$ segments, and randomly sampling short snippets from each segment; these sparsely sampled snippets are then passed through two-stream (spatial and temporal) Convolutional Neural Networks, and fused to derive a video-level prediction.

**I3D:** We utilize a 3-Dimensional action recognition model, I3D[1]. I3D borrows designs from 2-Dimensional networks, and inflate all filters and pooling kernels with an extra dimension. Although I3D also benefits from 2-Dimensional counterpart's learned parameters, we train our I3D models from scratch for consistency with other architecture experiments.

**R(2+1)D:** R(2+1)D [8] network offers (2+1)D convolution blocks which decomposes a 3D convolution block into a 2D convolution followed by a 1D convolution, aiming to model spatial and temporal information separately. This separation increases complexity of the model (due to the extra non-linearity between 2D and 1D convolution blocks), while having the same number of model parameters. Similar to [2], we keep the original output channel size rather than expanding it in order to directly load the ImageNet pre-trained weights.

## B    Representation Bias using I3D and R(2+1)D

Table S1: Representation bias computed using I3D. LP stands for linear probing.

| | Transferred Dataset | | | | | |
|---|---|---|---|---|---|---|
| | UCF101 | HMDB51 | Mini-SSV2 | Diving48 | IkeaFA | UAV-Human |
| ImageNet LP Accuracy, $\mathcal{M}(\mathcal{D}, \phi)$ | 59.13 | 34.51 | 12.07 | 9.59 | 34.15 | 2.65 |
| Representation Bias, $\mathcal{B}(\mathcal{D}, \phi)$ | 5.90 | 4.14 | 3.39 | 2.20 | 2.03 | 2.04 |

Table S2: Representation bias computed using R(2+1)D. LP stands for linear probing.

| | Transferred Dataset | | | | | |
|---|---|---|---|---|---|---|
| | UCF101 | HMDB51 | Mini-SSV2 | Diving48 | IkeaFA | UAV-Human |
| ImageNet LP Accuracy, $\mathcal{M}(\mathcal{D}, \phi)$ | 62.45 | 39.54 | 12.83 | 10.13 | 32.93 | 2.47 |
| Representation Bias, $\mathcal{B}(\mathcal{D}, \phi)$ | 5.98 | 4.33 | 3.48 | 2.28 | 1.98 | 1.94 |

We calculate representation bias for each downstream task again using I3D (shown in Table S1) and R(2+1)D (shown in Table S2). We show that there is no significant difference in representation bias for each downstream task, and therefore, bias values are not dependent on models.

36th Conference on Neural Information Processing Systems (NeurIPS 2022) Track on Datasets and Benchmarks.

# C    Additional Sensitivity Analysis: Effect of Number of Classes

Similar class sensitivity analysis can be made with I3D (Figure S1) and R(2+1)D (Figure S2).

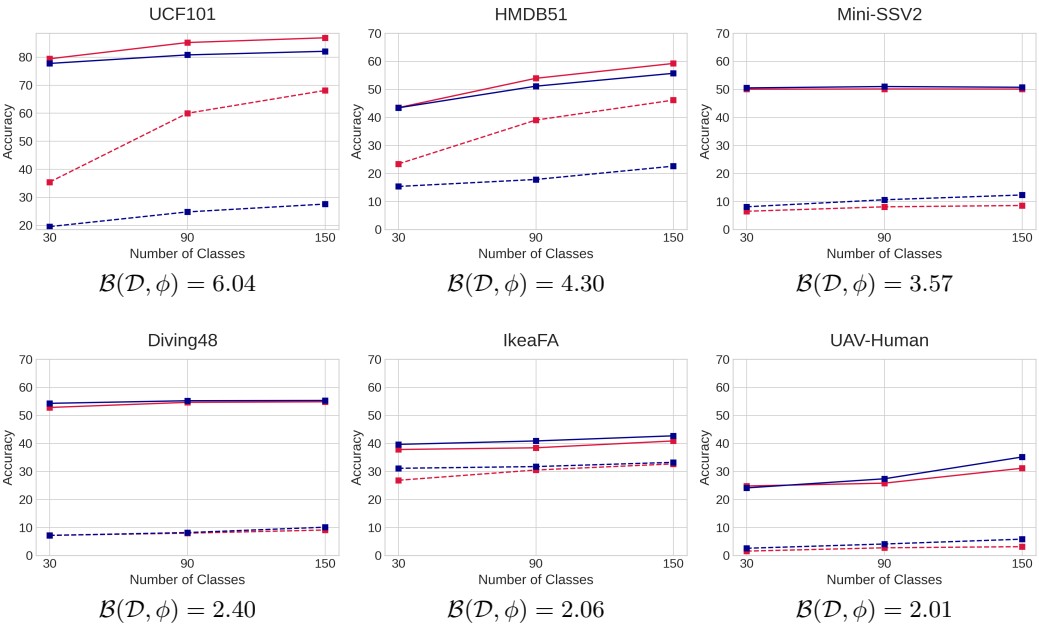

Figure S1: —— Kinetics FT; - - - Kinetics LP; —— Synthetic FT; - - - Synthetic LP.
**I3D** Fine-tuning (FT) and Linear Probing (LP) transfer results on six downstream tasks with various number of classes in pre-training datasets.

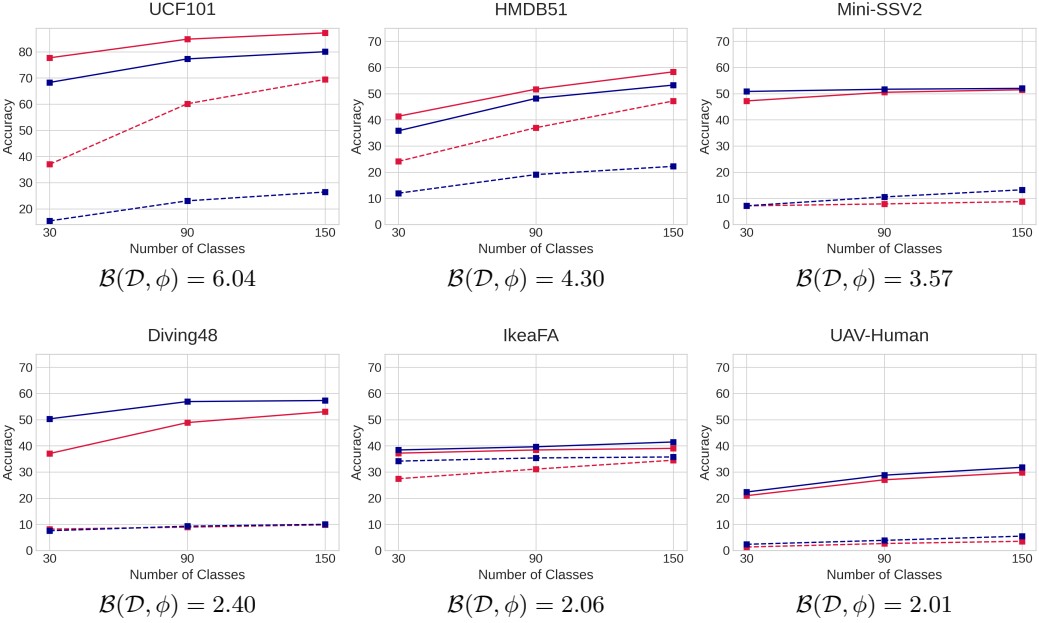

Figure S2: —— Kinetics FT; - - - Kinetics LP; —— Synthetic FT; - - - Synthetic LP.
**R(2+1)D** Fine-tuning (FT) and Linear Probing (LP) transfer results on six downstream tasks with various number of classes in pre-training datasets.

# D   Additional Sensitivity Analysis: Effect of Samples per Class

Similar sample sensitivity analysis can be made with I3D (Figure S3) and R(2+1)D (Figure S4).

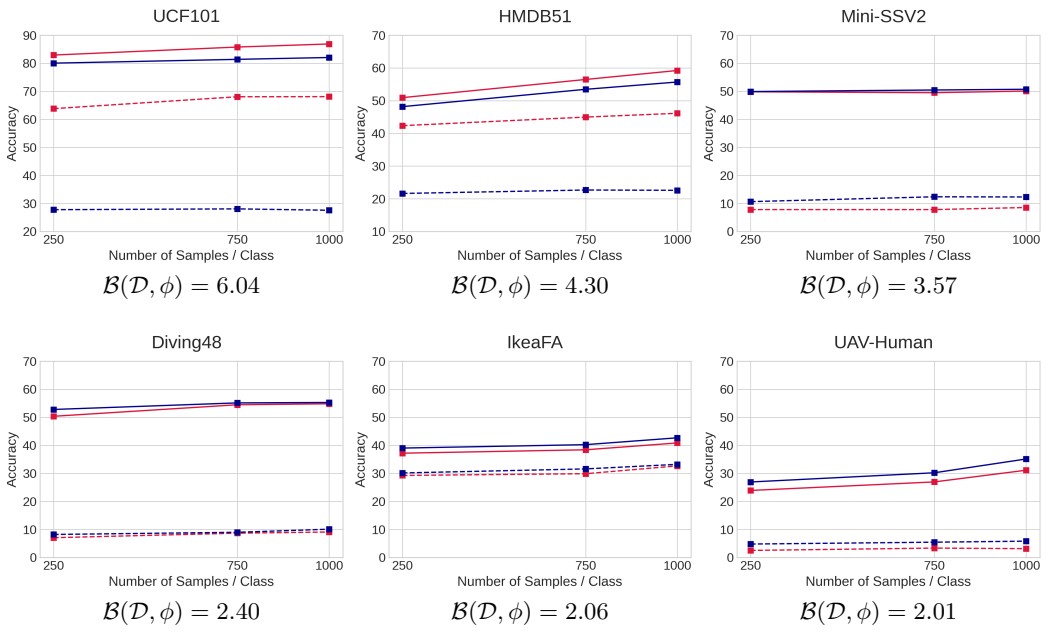

Figure S3: —— Kinetics FT; - - - Kinetics LP; —— Synthetic FT; - - - Synthetic LP.
**I3D** Fine-tuning (FT) and Linear Probing (LP) transfer results on six downstream tasks with various number of classes in pre-training datasets.

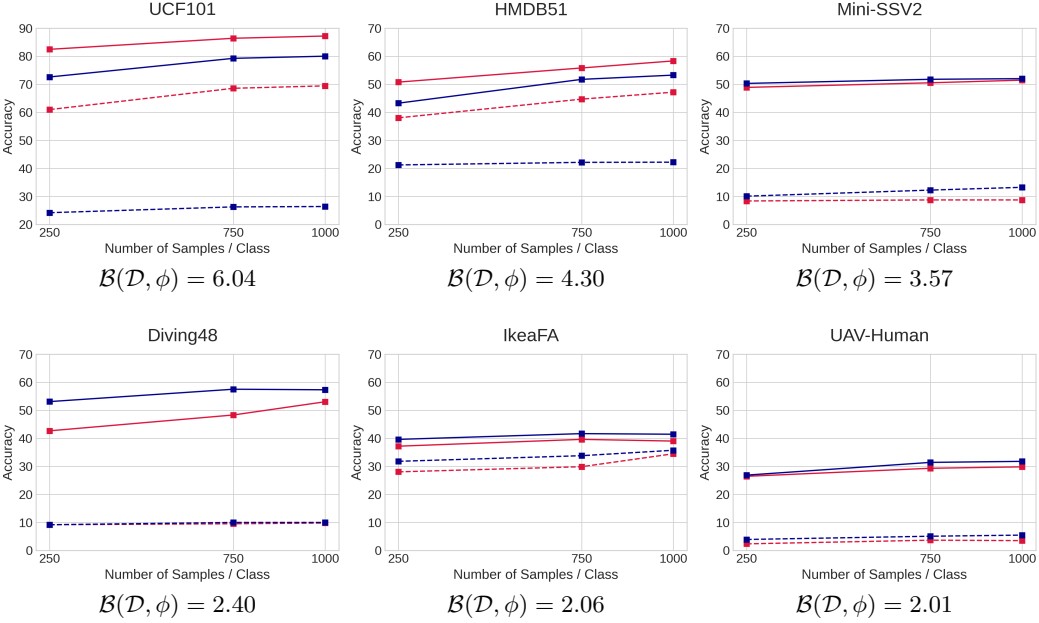

Figure S4: —— Kinetics FT; - - - Kinetics LP; —— Synthetic FT; - - - Synthetic LP.
**R(2+1)D** Fine-tuning (FT) and Linear Probing (LP) transfer results on six downstream tasks with various number of classes in pre-training datasets.

# E   Synthetic Dataset Frame Snapshots

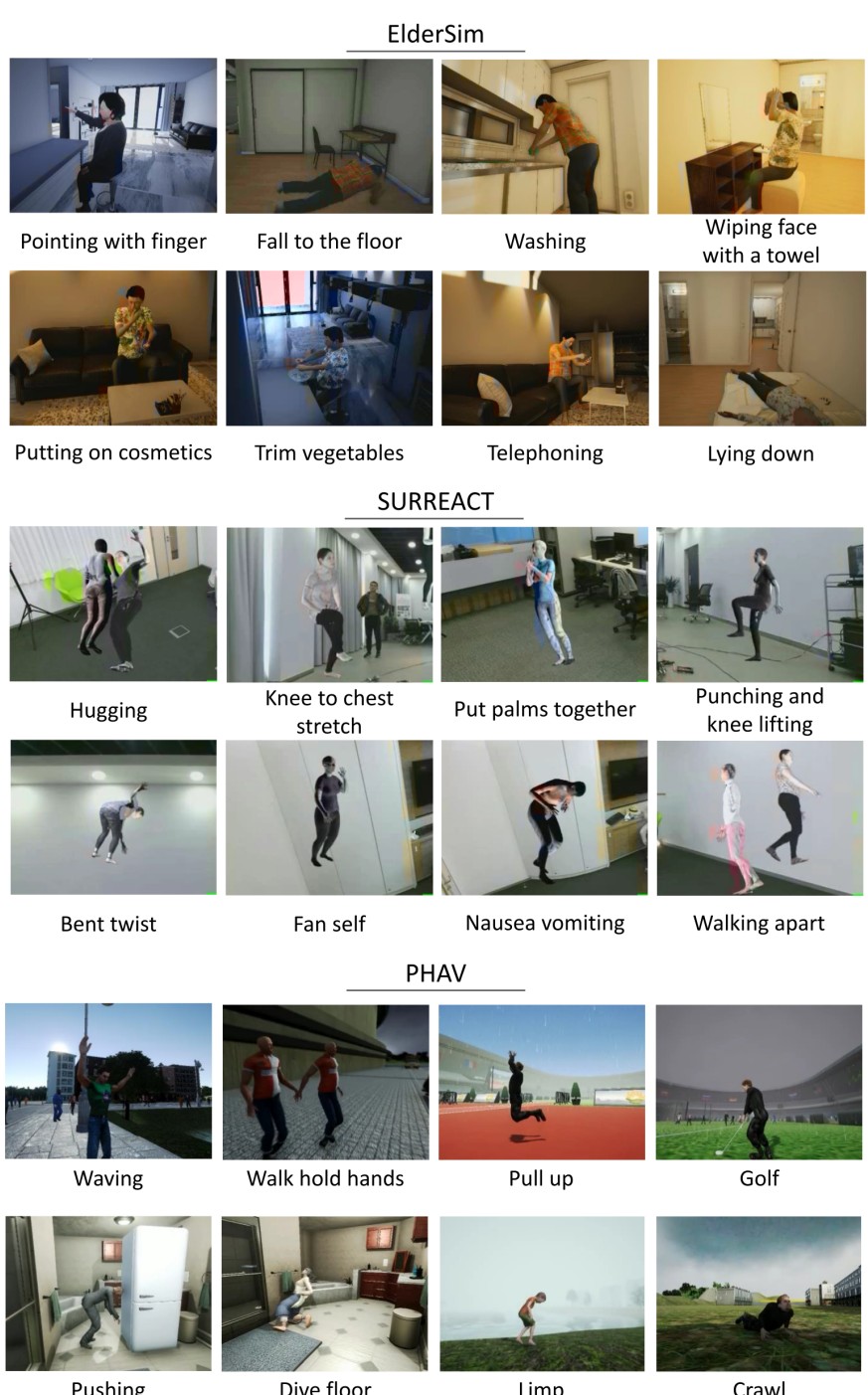

Figure S5: Examples of synthetic videos rendered by various simulators.

Figure S5 shows some synthetic videos and action categories used in our work. We emphasize that synthetic datasets also cover action categories, such as "falling to the floor", which are not easy to obtain from the real datasets.

## F    Downstream Tasks Statistics and Overlapping Classes

Table S3: Dataset statistics of downstream tasks.

| Datasets | # of Videos | # of Actions | Video Source | Domain |
|---|---|---|---|---|
| UCF101 [6] | 13,320 | 101 | YouTube | General |
| HMDB51 [3] | 6,849 | 51 | Movies/YouTube | General |
| Mini-SSV2 [2] | 93,000 | 87 | User-Provided | General |
| Diving48 [5] | 18,404 | 48 | Web | Diving |
| IkeaFA [7] | 111 | 12 | Self-collected | Assembly |
| UAV-Human [4] | 22,476 | 155 | Flying UAV | General |

Table S4: Summary of overlapping classes between pre-train Kinetics/Synthetic datasets and downstream tasks.

| Pre-trained Dataset | Transferred Dataset | | | | | | | | | | | |
|---|---|---|---|---|---|---|---|---|---|---|---|---|
| | UCF101 | | HMDB51 | | Mini-SSV2 | | Diving48 | | IkeaFA | | UAV-Human | |
| | # of classes | Ratio | # of classes | Ratio | # of classes | Ratio | # of classes | Ratio | # of classes | Ratio | # of classes | Ratio |
| Kinetics | 23 | 0.23 | 11 | 0.22 | 0 | 0.00 | 0 | 0.00 | 0 | 0.00 | 27 | 0.17 |
| Synthetic | 13 | 0.13 | 25 | 0.49 | 0 | 0.00 | 0 | 0.00 | 0 | 0.00 | 36 | 0.23 |

Table S3 shows detailed statistics of the six downstream tasks, and Table S4 summarizes the number of overlapping classes between Kinetics or Synthetic pre-train dataset and each of the six downstream tasks. Notice that Mini-SSV2, Diving48, and IkeaFA have completely disjoint action labels, and models pre-trained on Synthetic dataset outperform their respective Kinetics pre-trained models in these three datasets.

Interestingly, for HMDB51, the Synthetic pre-train dataset has more overlapping classes, yet the Kinetics pre-trained model still outperforms on this downstream task. Here, we conclude that the intersection of action labels plays a less significant role than representation bias.

## G    Computation Power

We ran experiments in parallel across various servers with 4, 6, or 8 GPUs; we used Titan V (12GB), Titan Xp (12GB), RTX 2080 (12GB), Titan RTX (24GB), and Tesla V100 (32GB).