# OpenReview forum: "How Transferable are Video Representations Based on Synthetic Data?"
_NeurIPS.cc/2022/Track/Datasets_and_Benchmarks — NeurIPS 2022 Datasets and Benchmarks _

### Official Review · Reviewer_JBRu · 2022-07-24
**Interesting study but missing some key evaluations**

**Rating:** 7
**Confidence:** 4
**Correctness:** Correct
**Clarity:** Yes, this paper is well written.

**Strengths:**

1.	This paper is the first to systematically study the pre-training potential of synthetic data on action recognition.
2.	Extensive experiments have shown that the synthetic dataset can work well on most downstream datasets, compared with Kinetics dataset.
3.	This paper is well written.


**Weaknesses:**

1.	Missing more ablation studies. AFAIK, the most current video action recognition works sample 16/32 frames for each clip, while the authors sample 8 frames in this paper. The authors are suggested to conduct experiments on 16 frames.
2.	Missing the experiments on transformer-based models, AFAIK, most current state-of-the-art models are transformer-based (e.g, TimeSformer, vivit, MViT). So, it is important to conduct experiments on such models.


**Additional Feedback:**

It is better to provide which 150 classes in Kinetics dataset are chosen, this will help the follower who will use this benchmark.

**Documentation:**

yes

**Relation To Prior Work:**

Yes, clear relationship to prior work is discussed.

**Summary And Contributions:**

This paper presents a systematic study on the pre-training potential of synthetic data on action recognition. A new action recognition benchmark (synthetic data) is proposed to mitigate the privacy, ethics, and copyright issues caused by real videos. Based on the study, several analyses are conducted, which could inspire future studies on synthetic action recognition research.

---

> ### Author Response · Authors · 2022-08-17
> **Thank you for the comments and feedback. Please find our initial responses below.**
>
> **Weaknesses-1 :**
> > Experiments with 16 frame baseline model
>
> We experimented with a TSN model using 16 frames (and FT for downstream tasks), and following are the results.
>
> |                           |  **UCF101**  |  **HMDB51**  |  **Mini-SSV2** | **Diving48** |  **IkeaFA**  |  **UAV-Human**
> |---------------------------|--------------|--------------|----------------|--------------|--------------|----------------
> | **Kinetics Pretrained**   |    87.02     |    60.52     |     50.14      |    68.38     |    38.41     |     33.82
> |  **Synthetic Pretrained** |    86.10     |    59.61     |     53.68      |    70.30     |    40.24     |     40.51
>
> We find that the model performs better with 16 frames than with 8. More importantly, same as our results with 8 frames, we find the Kinetics pre-training helps the model perform better on downstream tasks with high representation bias, while Synthetic pre-training helps it perform better on tasks with lower bias.
>
> **Weaknesses-2 :**
> > Transformer-based baseline models
>
> We agree that transformer models are a valuable addition to the baselines on this benchmark. We ran pre-training and transfer experiments with TimeSformer [e], and following are the results.
>
> | **Pre-training**    | **Downstream eval** | **UCF101** | **HMDB51** | **Mini-SSV2** | **Diving48** | **Ikea furniture** | **UAV** |
> |---------------------|---------------------|:----------:|:----------:|:-------------:|:------------:|:------------------:|:-------:|
> | **Kinetics**        | **LP**              |    89.43   |    55.42   |     21.54     |     16.99    |        47.73       |   8.40  |
> |                     | **FT**              |    92.09   |    59.51   |     48.92     |     46.37    |        61.93       |  23.27  |
> | **Synthetic**       | **LP**              |    82.12   |    49.17   |     21.24     |     19.20    |        45.45       |  13.79  |
> |                     | **FT**              |    89.03   |    54.38   |     51.10     |     44.86    |        63.64       |  24.95  |
>
> Similar to other baseline methods, we can see the downstream performance on tasks with higher bias is better with Kinetics pre-training than with Synthetic pre-training and vice-versa for lower bias downstream tasks. We will update our paper with the above results.
>
> Although the results above are better than our baseline models based on resnet backbones, we note that prior to pre-training the TimeSformer model on Kinetics or Synthetic data, we initialized it with the weights of a ViT trained on Imagenet-21K, as is common in action recognition literature with the use of transformers [b, e]. This is unlike the other baseline methods which were all randomly initialized prior to pre-training. We also attempted to train the TimeSformer model from scratch/random initialization, but were unable to train a good enough representation with the limited number of hyperparameter settings we could try, and consequently the downstream accuracies were much poorer and not representative of the baseline.
>
> **References**
>
> [e] Bertasius, Gedas, Heng Wang, and Lorenzo Torresani. "Is space-time attention all you need for video understanding?." ICML. Vol. 2. No. 3. 2021.

---

### Official Review · Reviewer_TbKx · 2022-07-25
**Introduces SynAPT, a benchmark for pre-training action recognition models from existing synthetic datasets with extensive analysis**

**Rating:** 7
**Confidence:** 5
**Correctness:** overall correct. some remarks are lis…
**Clarity:** very clear. enjoyed reading the analy…

**Strengths:**

1. Interesting analysis on the strength of pre-training using synthetic datasets vs real datasets from the downstream action recognition datasets. Results show models pre-trained on synthetic dataset performing equally well or outperforming the models pre-trained on kinetics when the datasets have low scene-object bias. A variety of downstream tasks are considered.
2. The downstream tasks that follow the pre-training have disjoint label sets. This shows that the SynAPT pre-training could be useful for a wider range of downstream tasks as there is no label constraint.
3. Interesting insights are provided into the effect of representation bias on the transferability and of number of classes on the performance.

**Weaknesses:**

1. For the sake of completeness, it might be interesting to report if similar/different results are observed in the transfer learning for the gap between the synthetic and real dataset pre-training on datasets with the same label set.
Further, since the label set is different, it might be nice to report a comparison with no pre-training. This would solidify that the pre-training indeed helped improve the results in the first place.

2. Is the choice of the real dataset (kinetics) well balanced with the choice of the combination of synthetic datasets? Are there differences between the two datasets beyond them being real/synthetic that could be biasing results? Would choosing a different "real" dataset affect the result?


**Additional Feedback:**

listed in weakness section.

**Documentation:**

code, datasets will be available upon publication according to the paper.

**Ethics:**

no.

**Relation To Prior Work:**

clear.

**Summary And Contributions:**

The paper uses existing synthetic datasets to pre-train models and then transfer to large scale action recognition datasets with disjoint label sets. The different datasets considered have a range of scene-object bias with the synthetic pre-training outperforming (or performing equally well wrt) the kinetics dataset pre-training on low scene-object bias datasets.

---

> ### Author Response · Authors · 2022-08-17
> **Thank you for the comments and feedback. Please find our initial responses below.**
>
> **Weaknesses-1 :**
> > Downstream tasks with the same label set as pre-training.
>
> The construction for such an experiment requires having the same label set in both synthetic and real pre-training data, which is unavailable in SynAPT. While subsets of our current benchmark can be created and correspondingly downstream tasks can be created/found, we leave this exploration to future work.
>
> > Results for downstream finetuning from scratch. Does pre-training help?
>
> In Tab 1, rows titled “Scratch” report accuracies of models trained on the downstream tasks from random initialization. They show that pre-training (with either real or synthetic data) helps downstream performance significantly.
>
> **Weaknesses-2 :**
> > Do the conclusions change if real pre-training data changes?
>
> To answer this, we conducted experiments pre-training a TimeSformer [e] model with a 150 class subset of the Moments in time dataset [f]. The results are reported in the table below.
>
> | **Pre-training**    | **Downstream eval** | **UCF101** | **HMDB51** | **Mini-SSV2** | **Diving48** | **Ikea furniture** | **UAV** |
> |---------------------|---------------------|:----------:|:----------:|:-------------:|:------------:|:------------------:|:-------:|
> | **Kinetics**        | **LP**              |    89.43   |    55.42   |     21.54     |     16.99    |        47.73       |   8.40  |
> |                     | **FT**              |    92.09   |    59.51   |     48.92     |     46.37    |        61.93       |  23.27  |
> | **Moments-in-time** | **LP**              |    87.95   |    52.71   |     20.27     |     14.87    |        42.05       |   7.15  |
> |                     | **FT**              |    91.24   |    57.01   |     48.15     |     45.16    |        46.59       |  21.67  |
> | **Synthetic**       | **LP**              |    82.12   |    49.17   |     21.24     |     19.20    |        45.45       |  13.79  |
> |                     | **FT**              |    89.03   |    54.38   |     51.10     |     44.86    |        63.64       |  24.95  |
>
> We see that similar to Kinetics, the accuracy of a model pre-trained with this Moments-in-time subset is higher than synthetic pre-training on tasks with higher representation bias, while it is lower on tasks with lower representation bias.
>
> **References**
>
> [e] Bertasius, Gedas, Heng Wang, and Lorenzo Torresani. "Is space-time attention all you need for video understanding?." ICML. Vol. 2. No. 3. 2021.
>
> [f] Monfort, Mathew, et al. "Moments in time dataset: one million videos for event understanding." IEEE transactions on pattern analysis and machine intelligence 42.2 (2019): 502-508.

---

> > ### Comment · Reviewer_TbKx · 2022-09-02
> > **Response of reviewer**
> >
> > Thank you for the authors' responses. My concerns have been addressed and rating stands at 'accept'.

---

### Official Review · Reviewer_TuGX · 2022-07-27
**An important work to study the transferability of synthetic dataset in action recognition but lacks some key elements**

**Rating:** 6
**Confidence:** 4

**Strengths:**

This paper explores a very important direction that we train with synthetic video datasets and test with real data. It can alleviate the privacy, ethical and data scarcity issues in ML world, especially for the action recognition task. The transferability of the pretrained models based on synthetic data is an important but under-explored topic for action recognition. The experimental results show that training with synthetic datasets can obtain more general video representations than the Kinetics dataset on several downstream classification tasks.

**Weaknesses:**

Though it is an important direction to explore the transferability of representation based on synthetic data overall, this paper lacks several key elements. There are several things to be clarified and provided:
1. The dataset documentation is missing. There is no detail for dataset construction, availability and maintenance in either Supplementary or web URL link. It makes the benchmark unreproducible.
2. The reason to choose the 150 classes for dataset construction is not clear. Are these classes randomly chosen from three synthetic datasets? or based on a specific standard. Although there is a sensitivity study in Chapter 4.3, the plots in Figure 3 show that the accuracies are not saturated when the classes increase from 30 to 150.
3. The representation bias of the proposed SynAPT and Kinetics are missing. Table 2 shows the bias of the downstream classification datasets based on ImageNet-pretrained model. However, the bias of training datasets including SynAPT and Kinetics is missing. The relationship between the bias of training data and test data is ignored, which makes the conclusion a bit weak.
4. Why the Kinetics is used as the representative of the 'real dataset'? To my knowledge, Kinetics is composed of public Youtube video dataset, which has similar size as SSV2 dataset. It is not surprising that Kinetics video representation can transfer better to HMDB51 and UCF101, because these 2 small datasets are fully/partially extracted from Youtube as well.
5. The transformer-based deep learning models are missing. The transformers like Timesformer, Swin, Mvit demonstrate superior performance on action recognition tasks in recent years.

**Additional Feedback:**

I would suggest the authors use more datasets for pre-training, to improve the generalizability of the hypothesis. At the same, exploring the relationship of the bias between training and test data.

**Clarity:**

There needs improvement in the writing. Table 1 shows the most important results in this paper, but the analysis based on it is little. It loses the connection with the assumption in the last paragraph of Chapter 4.2.

The motivation for only using Kinetics for comparison and the construction of the dataset needs to be further clarified.

**Correctness:**

There is no documentation of dataset construction and the chosen classes are not well explained. Hence, the construction of the current dataset is problematic.

Using the Kinetics dataset as the only 'real dataset' to compare with the synthetic dataset can make the hypothesis/conclusion weak. Though it is the most popular dataset in action recognition, we cannot get a general conclusion with such a single dataset.

**Documentation:**

There is no documentation provided for the dataset. The author mentioned in the paper that they will provide the details soon in the future.

**Ethics:**

N.A

**Relation To Prior Work:**

The discussion on prior work is sufficient.

**Summary And Contributions:**

This paper constructed a benchmark dataset SynAPT by combining the existing synthetic datasets. They pre-trained different CNN-based models with the benchmark dataset to evaluate the transferability of the video representation based on synthetic data. By comparing the results obtained from real datasets like Kinetics, they hypothesize that the performance gap from representation between the real and synthetic datasets is highly related to contextual bias and static objects. They finally conduct an ablation study to show the impact of the hyperparameters in dataset construction.

---

> ### Author Response · Authors · 2022-08-17
> **Thank you for the comments and feedback. Please find our initial responses below.**
>
> **Weaknesses-1, Correctness-1, Documentation :** We have released the dataset and documentation at [https://github.com/mintjohnkim/SynAPT](https://github.com/mintjohnkim/SynAPT), and updated the link in the revision.
>
> **Weaknesses-2 :**
> > Why 150 classes?
>
> 150 is the total number of classes (rounded to closest 10s) we could get from the synthetic assets. For downsizing Kinetics, we randomly picked 150 classes from it (the specific classes chosen have been mentioned in the benchmark documentation).
>
> **Weaknesses-3 :**
> > Representation biases of pre-training datasets
>
> Kinetics (150 classes) Representation bias = 6.07 (44.67% ImageNet LP Accuracy)
> Synthetic Representation bias = 5.07 (22.32% ImageNet LP Accuracy)
>
> As expected, representation bias of the Kinetics dataset is indeed significantly higher than that of synthetic data. Putting in perspective these numbers with results from Tabs 1, 2, we find that pre-training on a dataset with lower bias tends to do better on downstream tasks with lower bias.
>
> We have included the representation biases of the pre-training datasets in Table 2 of the revision.
>
> **Weaknesses-4, Correctness-2, Clarity, Additional Feedback :**
> > Kinetics as the representative ‘real dataset’
>
> In action recognition literature, Kinetics has been the go-to dataset for pre-training representations [b, c, d]. This was the motivation behind our choice.
>
> > More real pre-training datasets for analysis
>
> We agree that including more real datasets in the analysis can make our conclusions stronger, and so we conducted experiments pre-training a TimeSformer [e] model with a 150 class subset of the Moments in time dataset [f]. The results are reported in the table below (under the response for Weaknesses-5). We see that similar to Kinetics, the accuracy of a model pre-trained with this Moments-in-time subset is higher than synthetic pre-training on tasks with higher representation bias, while it is lower on tasks with lower representation bias.
>
> **Weaknesses-5 :**
> > Baseline models with transformer architectures
>
> We agree that transformer models are a valuable addition to the baselines on this benchmark. We ran pre-training and transfer experiments with TimeSformer [e], and following are the results.
>
> | **Pre-training**    | **Downstream eval** | **UCF101** | **HMDB51** | **Mini-SSV2** | **Diving48** | **Ikea furniture** | **UAV** |
> |---------------------|---------------------|:----------:|:----------:|:-------------:|:------------:|:------------------:|:-------:|
> | **Kinetics**        | **LP**              |    89.43   |    55.42   |     21.54     |     16.99    |        47.73       |   8.40  |
> |                     | **FT**              |    92.09   |    59.51   |     48.92     |     46.37    |        61.93       |  23.27  |
> | **Moments-in-time** | **LP**              |    87.95   |    52.71   |     20.27     |     14.87    |        42.05       |   7.15  |
> |                     | **FT**              |    91.24   |    57.01   |     48.15     |     45.16    |        46.59       |  21.67  |
> | **Synthetic**       | **LP**              |    82.12   |    49.17   |     21.24     |     19.20    |        45.45       |  13.79  |
> |                     | **FT**              |    89.03   |    54.38   |     51.10     |     44.86    |        63.64       |  24.95  |
>
> Similar to other baseline methods, we can see the downstream performance on tasks with higher bias is better with Kinetics pre-training than with Synthetic pre-training and vice-versa for lower bias downstream tasks. We will update our paper with the above results.
>
> Although the results above are better than our baseline models based on resnet backbones, we note that prior to pre-training the TimeSformer model on Kinetics or Synthetic data, we initialized it with the weights of a ViT trained on Imagenet-21K, as is common in action recognition literature with the use of transformers [b, e]. This is unlike the other baseline methods which were all randomly initialized prior to pre-training. We also attempted to train the TimeSformer model from scratch/random initialization, but were unable to train a good enough representation with the limited number of hyperparameter settings we could try, and consequently the downstream accuracies were much poorer and not representative of the baseline.
>
> **Clarity :**
> > Analysis of table 1
>
> Sec 4.2 is dedicated to the analysis based on Tab 1. Please let us know if there are additional questions.
> > Lost connection with assumption in Sec 4.2
>
> There is no assumption stated in the last paragraph of Sec 4.2. Could you please clarify the question?

---

> > ### Author Response · Authors · 2022-08-17
> > **References**
> >
> >
> > [b] Fan, Haoqi, et al. "Multiscale vision transformers." Proceedings of the IEEE/CVF International Conference on Computer Vision. 2021.
> >
> > [c] Wei, Chen, et al. "Masked feature prediction for self-supervised visual pre-training." Proceedings of the IEEE/CVF Conference on Computer Vision and Pattern Recognition. 2022.
> >
> > [d] Wang, Rui, et al. "Bevt: Bert pretraining of video transformers." Proceedings of the IEEE/CVF Conference on Computer Vision and Pattern Recognition. 2022.
> >
> > [e] Bertasius, Gedas, Heng Wang, and Lorenzo Torresani. "Is space-time attention all you need for video understanding?." ICML. Vol. 2. No. 3. 2021.
> >
> > [f] Monfort, Mathew, et al. "Moments in time dataset: one million videos for event understanding." IEEE transactions on pattern analysis and machine intelligence 42.2 (2019): 502-508.

---

> > ### Comment · Reviewer_TuGX · 2022-08-21
> > **Thanks for the fast response**
> >
> > Thanks for the responses from the authors. I think the Github link and the extra experiments have addressed my concerns well. I will raise my rating to acceptance. I am also looking forward to the revised version of the paper. I agree that synthetic training data is an important direction that deserves more exploration.

---

### Official Review · Reviewer_4wDV · 2022-07-27
**Review for the SynAPT benchmark**

**Rating:** 7
**Confidence:** 3
**Correctness:** 1. Points to an important challenge w…
**Clarity:** The paper is well-written and well-or…

**Strengths:**

1. SynAPT provides a useful benchmark for the action recognition community, while minimizing issues caused by ethical concerns and object/scene bias of real videos
2. The extensive study of the sim-to-real gap in action recognition addresses important challenges in the research community
3. The experimental design, including choice of evaluation dataset and model architecture, is well thought out and shows an understanding of the field.

**Weaknesses:**

1. For some dataset (especially UAV-human), linear probing accuracy is extremely low for both real and synthetic datasets, and performance is even lower for the "scratch" model. The 1.26 accuracy for R(2+1)D seems suspiciously low, even for a randomly initialized model, on a dataset with 22,476 video clips.
2. All backbones are initialized randomly, without ImageNet (Line 200), without justification. I would like to see at least the "scratch" model with ImageNet initialization.

**Additional Feedback:**

The authors should provide the script to generate the dataset. They should also provide evaluation code so that future researchers can fairly compare their models on SynAPT.

**Documentation:**

SynAPT is constructed from existing datasets, which are separately available, using a script to construct SynAPT from the existing datasets. The authors claim that they "will provide the respective scripts for the community to generate this synthetic data" (Line 162), although no script is provided or linked to.

In the supplementary materials, the authors provide further details about the model architecture and training.

**Ethics:**

No - conversely, ethics is a motivating factor in the construction of this synthetic data benchmark.

**Relation To Prior Work:**

Yes. Unlike previous works, the authors conduct a study of the transferability of learned representations from models trained on fully synthetic videos to downstream tasks. SynAPT is constructed from three previous synthetic datasets (ElderSim, SURREACT, and PHAV).

**Summary And Contributions:**

1. The authors introduce SynAPT, a novel benchmark of synthetic video clips compiled from existing datasets
2. The authors provide the first comprehensive study of the transferability of action representations based on synthetic videos to diverse downstream task, showing a close relationship between the sim-to-real gap and the object/scene bias
3. The authors address ethical concerns with existing datasets

---

> ### Author Response · Authors · 2022-08-17
> **Thank you for the comments and feedback. Please find our initial responses below.**
>
> **Weaknesses-1:**
> > Low accuracy on UAV dataset
>
> UAV is a highly challenging dataset of unmanned aerial vehicle videos with 155 action classes occurring in a relatively small receptive field. This makes it hard for a randomly initialized model to perform well on these videos.
>
> **Weaknesses-2:**
> > Pre-training without Imagenet initialization
>
> Imagenet consists of real images and hence pre-training on it carries along the same costs and concerns as pre-training with real videos (e.g. privacy, ethical or copyright concerns). In our study, this is precisely what we wish to avoid, and hence we pre-trained using synthetic videos from random initialization.
>
> **Correctness-1:**
> > ln45 comment about same label sets in prior use of synthetic videos for action recognition
>
> In the statement mentioned, we tried to distinguish our work from recent action recognition work using synthetic data (ln 43-44), and the ln45 statement is specifically about techniques used in those papers. To the best of our knowledge, ours is the first to explore pre-training of action recognition models on synthetic data and subsequent transfer to a range of different downstream tasks that do not have the same label set as pre-training.
>
> In the papers mentioned in the reviewer comment, unsupervised pre-training is not based on synthetic data and is not transferred to multiple different downstream tasks, making them distinct from our work.
>
> **Documentation:** We have released the scripts and documentation for our dataset at [https://github.com/mintjohnkim/SynAPT](https://github.com/mintjohnkim/SynAPT) and included the link in the revision.

---

### Official Review · Reviewer_usaV · 2022-07-28
**Benchmark based on compiled datasets shows the utility of synthetic data for pre-training in video action recognition**

**Rating:** 6
**Confidence:** 3
**Correctness:** Yes.
**Clarity:** Yes.

**Strengths:**

(+) The paper proposes a compiled and curated version of multiple synthetic video action recognition datasets, by combining classes, normalizing the frame rate, or balancing the number of samples per class.

(+) The paper provides a valuable list of advantages of synthetic datasets, discussing privacy and ethics issues.

**Weaknesses:**

(-) Motivation for not reporting state-of-the-art performance not convincing
The paper only reports results on a pre-training with a subset of Kinetics. The authors justify this with an emphasis on fair comparisons between synthetic and realistic pre-trained models. Still, it would have been interesting to know what is the empirical upper bound is at the moment and what is needed to bridge this performance gap (e.g., is it more samples? better rendered videos? more diverse videos?)

(-) Unclear goal of the benchmark given the evaluated method
* If the emphasis of the paper is to compare synthetic and realistic pre-trained models, how come there is no domain adaptation methods that were assessed?
* On related topic, it is hard to grasp the main message of the benchmark. What would the authors recommend for future works working on such benchmark? e.g., should the models focus on better leveraging synthetic data, better transferring knowledge, mitigating dataset biases?

(-) Biases metric scores in Table 2 would benefit from a more thorough analysis
It is a bit hard to understand and interpret the scores. For example, Mini-SSV2 has a representation bias score very close to HMDB51 but is considered as focusing more on temporal movement. Similarly, IkeaFA has a very high performance with ImageNet backbone but is considered as focusing more on temporal movement. What is the explanation for these two datasets contradictory?

(-) Statements can be overclaimed
It would be preferable to avoid terms like "theorized" as the paper does not propose any theory but rather some hypothesis on why some datasets might benefit more from a synthetic training.

**Additional Feedback:**

n.a.

**Documentation:**

Not at the moment, but the author promised code availability to reproduce the compiled dataset.

**Ethics:**

No.

**Relation To Prior Work:**

Yes.

**Summary And Contributions:**

The paper curates existing synthetic datasets to compile a new dataset to benchmark video action recognition.
When using the resulting dataset for pre-training action recognition models, it appears that synthetic pre-training benefits the performance on real datasets differently.
The paper explores the idea of a bias towards objects behind these performance discrepancy.

---

> ### Author Response · Authors · 2022-08-17
> **Thank you for the comments and feedback. Please find our initial responses below.**
>
> **Weaknesses-1 :**
> > Full Kinetics results as upper bound
>
> We agree that the results with a full Kinetics pretrained model can be reported as an upper bound and will include them in the final version of the paper. We currently have run experiments to get the results for UCF101 and HMDB51, with TSN and I3D models (8 frames) with full-network downstream finetuning. We note here that prior to full Kinetics pre-training, the models were initialized with the weights of a Resnet-50 network trained on Imagenet-1K classification, while our synthetic pretrained models were initialized with random initialization..
>
> Numbers in parentheses are with synthetic data pre-training.
>
> |                    | **UCF101**   | **HMDB51**   |
> |--------------------|--------------|--------------|
> | **TSN (ResNet50)** | 90.2 (83.40) | 62.5 (54.38) |
> | **I3D (ResNet50)** | 89.9 (82.05) | 60.2 (55.69) |
>
> > Is it more samples that can help?
>
> In Sec 4.3, we conducted some analyses with a different number of classes (Fig 3) and a different number of samples per class (Fig 4). We observed that adding more classes and samples helps improve performance in most downstream datasets, albeit at different rates. Extrapolating optimistically, this may potentially be a factor that helps close the gap to the upper bound, and we hope to expand the benchmark in the future possibly with more synthetic datasets developed in-house or contributed by the community.
>
> **Weaknesses-2:**
> > Goal of the benchmark
>
> The goal of the benchmark is to create a standardized comparison between real and synthetic pre-training for downstream action recognition. This benchmark can then support different pre-training and transfer methods for performance comparison between real and synthetic data pre-training. It is also open to contributions for expanding the synthetic dataset.
>
> > Domain adaptation methods for evaluation
>
> Most research in domain adaptation often assumes a label set overlap between source and target, while in our benchmark, all downstream tasks are distinct and do not fully overlap in class labels with pre-training data. Although techniques from domain adaptation literature may find use in this scenario, we leave the testing of such methods on our benchmark to future work (possibly from the development of new methods using our benchmark).
>
> > Main messages of the benchmark
>
> One of the main messages of the paper is that pre-training with synthetic data overcomes different costs and concerns (ethical, privacy, or copyright related) associated with using real data. Secondly, using the results of multiple baseline methods we analyzed relationships between performance of synthetic pre-training and a dataset’s representation bias.
>
> While future methods on the benchmark can improve upon baseline downstream performances, it might also be interesting to see if any, with existing data, exhibit different behavior to the baselines and perform better with synthetic pre-training. Future works on this benchmark can also contribute more synthetic data and study the behavior of baseline methods in this scenario.
>
>
> **Weaknesses-3:**
> > HMDB51 and Mini-SSV2 have different characteristics but similar bias
>
> Note that the representation bias is calculated on a logarithmic scale as shown in Equation (1) of Sec 4.2. Therefore, the gap between the two datasets (HMDB51: 4.30 and Mini-SSV2: 3.57) is significant.
>
> > IkeaFA has high Imagenet LP performance but low bias
>
> IkeaFA’s perceived high accuracy results from it being a simpler classification task with 12 classes (8.33% chance accuracy). In the representation bias computation, this is factored in through the chance accuracy term ($\mathcal{M}_{rnd}$) in the denominator (Eq 1.). Hence, both IkeaFA and Mini-SSV2 have lower bias and are more temporal movement-focused compared to HMDB51.
>
> **Weaknesses-4:**
> > “theorized” should not be used
>
> Thank you. The change “theorized” -> “hypothesized” has been made in our revision.

---

> > ### Comment · Reviewer_usaV · 2022-08-27
> > **Comment to response**
> >
> > Thank you for addressing my concerns. I have increased my score accordingly.
> > It would be interesting to see in the future how the proposed benchmark evolves and what directions are taken to bridge the domain gap.

---

### Author Response · Authors · 2022-08-17
**Initial responses added under each reviewer's comments**

We thank the reviewers very much for their constructive comments. We are encouraged that reviewers found our paper addresses important challenges and alleviates many concerns with ethics, privacy and copyright by training with synthetic videos for action recognition (usaV, 4wDV, TuGX). And that the paper shows extensive experiments by considering multiple downstream tasks (4wDV, TbKx, JBRu). Moreover, reviewer TbKx agrees that one key advantage of SynAPT pre-training is being able to transfer representations to a wide range of downstream tasks without label constraints.

We added our initial responses including all clarification and additional experiments, under each reviewer’s main comment.

We added public documentation for benchmark reproducibility at: [https://github.com/mintjohnkim/SynAPT](https://github.com/mintjohnkim/SynAPT). Additionally, we included a [datasheet](https://github.com/mintjohnkim/SynAPT/blob/main/DataSheet.pdf) for our dataset, following Gebru et. al.’s work [a].

For addressing reviewer concerns, we conducted the following experiments:
- Added a baseline method with a transformer architecture in the form of TimeSformer to our benchmark as suggested by reviewers TuGX and JBRu.
- Conducted experiments where we pre-train with another real video dataset : Moments-in-time (as suggested by reviewers TuGX and TbKx).
- Conducted experiments with models using 16 frames (suggestion from reviewer JBRu)
- Added representation biases for pre-training datasets (as suggested by reviewer TuGX).

Hoping for a productive discussion!

**References**

[a] Gebru, Timnit, et al. "Datasheets for datasets." Communications of the ACM 64.12 (2021): 86-92.

---

### Meta-Review · Area_Chair_NHLw · 2022-09-03

**Recommendation:** Accept
**Confidence:** 5

**Metareview:**

This paper had 5 reviews. In response to the reviewers' comments, the authors performed additional experiments. Considering that the dataset consists of synthetic videos, there are no concerns related to ethics, privacy, and copyright. Public documentation for benchmark reproducibility is available on GitHub. Overall, the authors responded satisfactorily to the reviewers' comments, and all reviewers were satisfied with their response. The paper is well written and its relation to prior work are clear. The paper is original and of significance to deep learning and action recognition.

---

### Decision · Program_Chairs · 2022-09-16

Accept